# The *sprT* Gene of *Bacillus velezensis* FZB42 Is Involved in Biofilm Formation and Bacilysin Production

**DOI:** 10.3390/ijms242316815

**Published:** 2023-11-27

**Authors:** Yihan Yang, Ruofu Chen, Mati Ur Rahman, Chunyue Wei, Ben Fan

**Affiliations:** 1Co-Innovation Center for Sustainable Forestry in Southern China, College of Forestry, Nanjing Forestry University, Nanjing 210037, China; yihanyang@njfu.edu.cn; 2College of Life Science, Nanjing Forestry University, Nanjing 210037, China; crf60427@njfu.edu.cn (R.C.); mati@njfu.edu.cn (M.U.R.); chunyuewei@njfu.edu.cn (C.W.)

**Keywords:** *Bacillus velezensis* FZB42, *sprT*, biofilm, swarming, sporulation, bacilysin

## Abstract

*Bacillus velezensis* FZB42, a representative strain of plant-growth-promoting rhizobacteria (PGPR), can form robust biofilm and produce multiple antibiotics against a wild range of phytopathogens. In this study, we observed different biofilm morphology of the mutant Y4, derived from a TnYLB-1 transposon insertion library of *B. velezensis* FZB42. We identified that the transposon was inserted into the *sprT* gene in Y4. Our bioinformatics analysis revealed that the SprT protein is an unstable hydrophilic protein located in the cytoplasm. It is highly conserved in *Bacillus* species and predicted to function as a metalloprotease by binding zinc ions. We also demonstrated that Δ*sprT* significantly reduced the swarming ability of FZB42 by ~5-fold and sporulation capacity by ~25-fold. In addition, the antagonistic experiments showed that, compared to the wild type, the Δ*sprT* strain exhibited significantly reduced inhibition against *Staphylococcus aureus* ATCC-9144 and *Phytophthora sojae*, indicating that the inactivation of *sprT* led to decreased production of the antibiotic bacilysin. The HPLC-MS analysis confirmed that bacilysin was indeed decreased in the Δ*sprT* strain, and qPCR analysis revealed that Δ*sprT* down-regulated the expression of the genes for bacilysin biosynthesis. Our results suggest that the *sprT* gene plays a regulatory role in multiple characteristics of *B. velezensis* FZB42, including biofilm formation, swarming, sporulation, and antibiotic production.

## 1. Introduction

Many soil-borne phytopathogens can destroy plants or strongly reduce plant production, causing a significant economic loss in agriculture or forestry [1,2]. Although chemical pesticides can effectively control plant diseases, they often lead to ecological and environmental problems. In contrast, biological control strategies are currently receiving increasing attention due to their environmentally friendly nature. There are a large number of beneficial microorganisms in plant rhizosphere, which are generally defined as plant-growth-promoting microorganisms (PGPMs). These microorganisms have plant disease control functions and thus can be developed as biocontrol agents; meanwhile, most of them also facilitate plant growth and can be used as microbial fertilizers [3].

*Bacillus velezensis* FZB42 is a model organism of plant-growth-promoting rhizobacteria (PGPR). It can colonize plant roots [4,5] and synthesize a variety of secondary metabolites, hormones, and antioxidants, helping plants against phytopathogens [6,7,8]. FZB42 also induces systemic resistance of plants [9,10]. Nearly 10% of the FZB42 genome is used to synthesize antimicrobial metabolites and their corresponding immune genes [11,12]. Among them, five gene clusters encode non-ribosomal peptide synthases (NRPSs), which generate lipopeptide antibiotics (LPs), and three clusters encode polyketide synthase (PKS) genes, which produce polyketide (PKs) antibiotics [7,13,14,15]. A dipeptide antibiotic produced by FZB42, bacilysin, also has strong antimicrobial activity [16]. For example, bacilysin demonstrates remarkable inhibition of dangerous plant pathogens such as *Phytophthora* and *Xanthomonas* [2,17]. In addition, bacilysin strongly suppresses the growth of some eukaryotes such as *Saccharomyces cerevisiae* and *Candida albicans* [17,18].

The production of these antibiotics in *B. velezensis* FZB42 is well connected with other phenotypes such as biofilm formation, motility, and sporulation [19]. Unlike spores produced by fungi, *Bacillus* survives longer in adverse conditions by producing heat-resistant endospores [20]. In order to further understand how these characteristics are regulated in FZB42, we constructed a transposon library of FZB42 using the TnYLB-1 transposon [21]. Through observing colony morphologies of the mutant strains from the library, we identified a mutant strain (Y4) with a seemingly altered colony surface. In this study, we determined the disrupted gene in Y4 via TnYLB- 1 and studied its effect on biofilm formation, motility, and sporulation. Moreover, we explored the implication and the primary mechanism of this gene (*sprT*) in the production of the antibiotic bacilysin.

## 2. Results and Analysis

### 2.1. sprT Is Involved in the Biofilm Formation of B. velezensis FZB42

In a previous piece of work, we screened the TnYLB-1 transposon library of *B. velezensis* FZB42 [21] and noticed that the insertional mutant Y4 has a slightly different colony morphology (Figure 1), indicating that Y4 may be affected in biofilm formation. We examined the pellicle formation of Y4 in LB medium. The result showed that, after 34 h incubation, Y4 formed a less wrinkled pellicle than the wild-type FZB42. This confirmed that Y4 has a defected ability of forming biofilm. We analyzed the position of TnYLB-1 insertion and found it inserted in the *sprT* gene (Figure 2A). Since we were unable to transform a plasmid for *sprT* complementation in Y4, we independently constructed a new mutant by knocking out *sprT* using the homologous recombination method to avoid the possibility that other point mutations during transposition caused the phenotype change in Y14. The generated knockout strain was named Y14 and examined for its ability to form biofilm. We found that Y14 displayed a similar biofilm result to Y4 at either a solid–gas interface or a liquid–gas interface. These results confirmed that *sprT* should be responsible for the biofilm defects of Y4 and Y14.

### 2.2. Bioinformatic Analysis Shows sprT Is Conserved among Bacillus Species

Before deep exploration of the function of *sprT*, we performed a bioinformatic analysis. The gene was predicted to encode a polypeptide (SprT) consisting of 149 amino acids, with a molecular weight of 17.709 kDa, belonging to the metalloprotease family. The SprT protein has an isoelectric point of 9.84, a half-life of 30 h, and an instability index of 56.01, which is greater than the threshold value of 40. Its fat coefficient is 65.50, and its average hydrophilicity is −0.909 (negative value). Therefore, it is predicted that SprT is an unstable hydrophilic protein. No threonine phosphorylation site was predicted in SprT, but there were two phosphorylation sites (43 S and 59 S) on serine residues. No signaling peptide was predicted, indicating that SprT is a non-secretory protein.

In the SprT protein, the α-helix has 52 amino acids, accounting for 34.90%; the β-sheet has 12 amino acids, accounting for 8.05%; the random coil has 59 amino acids, accounting for 39.60%; and the extended chain has 26 amino acids, accounting for 17.45% (Figure 2B). Therefore, α-helix and random coils are the most abundant secondary structures in SprT. The amino acids 67H, 68E, and 71H located on one α-helix were predicted to be the active sites, and among which, 67H and 71H may have a zinc ion binding function (Figure 2B).

The sequence alignment analysis showed that the SprT protein sequence was conserved in the *Bacillus* species, and especially highly conserved at 67H, 68E, and 71H but less conserved in other phylogenetically distant species (Appendix A, Figure 2C). This indicates that the function of *sprT* may only be specific to the *Bacillus* species.

### 2.3. sprT Deletion Resulted in a Defect in Swarming Ability and Sporulation of FZB42

Motility and sporulation are closely linked to biofilm formation in *Bacillus*. To test whether *sprT* also affects the motility of FZB42, we examined their swarming ability on 0.5% LB solid medium. The mean diameter of colonies of the FZB42 wild type was 34.25 mm 12 h after incubation, while the mean diameters of Y4 and Y14 were 7.43 mm and 6.43 mm, respectively (Figure 3A,B), whereby both of which were significantly smaller than that of the wild type. When the FZB42 wild-type colonies reached 90 mm in diameter after 18 h, the mean diameters of Y4 and Y14 were 24.6 mm and 17.8 mm, respectively (Figure 3A,B). They were still smaller than those of the wild type. These results indicated that deletion of *sprT* reduced the swarming ability of FZB42.

To determine whether Δ*sprT* affects the spore formation of FZB42, we collected the cultures after 24 h of incubation of the assays. After heat treatment and five rounds of serial dilution, the endospores were spotted on plates for quantification. The results showed that the sporulation rate of the FZB42 wild type was maintained at the dilution of 5^−9^, which is approximately 25 times that of the sporulation rates of Y4 and Y14 (Figure 3C,D). The result indicated that *sprT* is also involved in the sporulation of FZB42.

### 2.4. sprT Deletion Reduces Production of the Antibiotic Bacilysin

In FZB42, regulations of antibiotic production, biofilm formation, and sporulation share some similarities. To test whether *sprT* is also involved in the antibiotic production of FZB42, we used *Staphylococcus aureus* ATCC-9144 and *Phytophthora sojae* as indicator strains, whereby both of which are sensitive to the antibiotic bacilysin [2,22]. The results showed that the sterile supernatants of the wild-type FZB42 cultures displayed strong antagonistic ability toward *S. aureus* ATCC-9144 and *P. sojae,* while the supernatants of the Δ*bacC* mutant (the negative control) and PA medium (the blank) had no inhibitory effect. However, the Δ*sprT* mutant demonstrated a significantly smaller inhibitory zone than the wild type against ATCC-9144 (Figure 4A,B). Similarly, the Δ*sprT* mutant showed a significantly reduced inhibitory ability against *P. sojae* (Figure 4A,C). Furthermore, we compared the yield of bacilysin in the strains using the HPLCMS method. The result showed that the wild-type strain produced 1.43 times the amount of bacilysin that Δ*sprT* produced (Figure 4D,E). Taken together, it can be concluded that *sprT* plays a positive regulatory role in the production of bacilysin.

### 2.5. Transcription of Bac Operon Is Down-Regulated via ΔsprT Deletion

The production of bacilysin in the Δ*sprT* mutant may result from lowered biosynthesis and exportation. To pinpoint which process is affected in the Δ*sprT* mutant, we first examined the transcription of the *bac* operon which encodes the synthetases for bacilysin production. The *bac* operon is composed of *bacA*, *bacB*, *bacC*, *bacD*, *bacE*, *bacF*, and *bacG* genes. Using qPCR, we found that almost all of the genes were significantly down-regulated in Δ*sprT*, except for the *bacA* gene. In particular, the transcript levels of the downstream parts of the *bac* operon, that is, *bacC*–*bacG*, were no more than half the level of what was expressed in the wild type (Figure 5). These results suggest that Δ*sprT* affects the production of bacilysin at the level of transcription.

## 3. Discussion

With a pyramid of previous studies, *B. velezensis* FZB42 has become one of the most well-studied PGPR strains [11,23]. In this work, we observed different biofilm morphologies of the mutant Y4 obtained from a transposon library. The insertion site of Y4 was determined to be the *sprT* gene. We found that the deletion of *sprT* had a negative effect on the biofilm formation, motility, sporulation, and antibiotic production of FZB42. In *Bacillus* spp., the SprT protein is predicted to be a metalloproteinase and is conserved. The association between metalloprotease and biofilm has been reported in pathogenic bacteria. For example, a serine alkaline metalloprotease from *Halobacillus karajensis* was effective for biofilm reduction in *Pseudomonas aeruginosa* and *S. aureus* [24]. In addition, a metalloprotease aureolysin from *Staphylococcus warneri* inhibited biofilms of *S. warneri*, and the aureolysin-inactivated mutant formed biofilm about twice as much as the wild type [25]. In contrast to the pathogenic bacteria, FZB42 is a plant-beneficial bacterium. Its effect of the *sprT* gene on biofilm and other related phenotypes provides new insights into the functions of metalloprotease.

Biofilm formation requires bacteria to first develop surface attachment, and studies have shown that motility is critical for initial attachment and biofilm formation [26]. Moreover, biofilms and motility share common regulation, for example, both biofilm formation and swarming motility are activated by low and very low levels of DegU~P, but inhibited by high levels of DegU~P [27]. We tested the swarming ability of ∆*sprT* and the results showed that the ∆*sprT* strain nearly lost its motility. This may be one reason for impaired biofilm formation of the *∆sprT* strain. To date, there has been little research on the link between metalloproteases and bacterial motility. One extracellular metalloprotease, Zmp1, has been found in *Clostridium difficile* and is involved in the regulation of adhesion motility [28]. The specific mechanism for this link remains unclear. The results we discovered here may provide a new way to study the relationship between bacterial motility and metalloproteases in the future.

Sporulation is closely linked to biofilm formation and motility in *Bacillus*. The research by Shankweiler found that loss of a neutral metalloprotease had no detectable effect on growth, morphology, or sporulation in *B. subtilis*, indicating that the metalloproteinase is not essential for the development of endospores [29]. However, our results showed that sporulation of FZB42 decreased by about 25 times after the deletion of *sprT*. Sporulation initiation in *B. subtilis* is controlled by the phosphorylated form of the master regulator Spo0A, which controls the transcription of a multitude of sporulation genes [30]. The metalloprotease InhA from *Bacillus thuringiensis* has been shown to be related to the sporulation gene *spo0A* [31]. We cannot exclude the possibility that there may be also a correlation between *spo0A* and sporulation.

In FZB42, biofilm formation, motility, and sporulation are connected with antibiotic synthesis, since they are regulated by a network with some shared regulators. Bacilysin, produced by FZB42, is a rather simple dipeptide composed of only one L-alanine and one unusual amino acid called anticapsin [32]. Our results showed that ∆*sprT* reduced the yield of bacilysin. The *bac* operon, a gene cluster responsible for bacilysin synthesis and secretion, was detected via qPCR, showing that almost all *bac* genes were significantly down-regulated. The effect of metalloprotease genes on the yield of bacilysin was reported for the first time in FZB42.

Heterotrophic bacteria usually secrete proteases, mainly serine proteases and metalloprotease, outside of their cells to degrade environmental proteins for nutrition purposes [33]. Therefore, we also believe that metalloprotease plays a role in providing substrates for the synthesis of antibiotics. The deletion of the metalloprotease gene *sprT* leads to a reduction in the raw material for antibiotic synthesis, which in turn reduces the production of bacilysin. On the other hand, we suggest that there is a relationship between motility and antibiotic synthesis in FZB42. We speculate that the deletion of the *sprT* gene first affects the motility of FZB42, causing some unknown changes, which in turn affects the yield of bacilysin [34,35]. However, further research is needed to test this hypothesis.

In summary, our results have demonstrated the importance of the metalloprotease gene *sprT* for biofilm formation, swarming, sporulation, and antibiotic production in FZB42. Understanding the role of this metalloprotease in the synthesis of FZB42 antibiotics can help in the use of PGPR strains, such as FZB42, to play a better role in biocontrol. In the future, in-depth studies can be performed to uncover the molecular mechanisms underlying these effects.

## 4. Materials and Methods

### 4.1. Strains, Growth Conditions, and Primers

The bacterial cells, both *S. aureus* and FZB42, were routinely grown in LB medium (Tryptone 10 g, yeast extract 5 g, NaCl 10 g [36]). FZB42 was grown in PA medium (KH_2_PO_4_ 1.1 g, MgSO_4_·7H_2_O 0.55 g, KCl 0.55 g, glutamic acid 3.828 g, sucrose 13.7 g, Na_3_C_6_H_5_O_7_·2H_2_O 0.1 g, FeCl_3_ 0.1 g) at 37 °C for bacilysin production. *P. sojae* was cultured on solid V8 agar medium [37] (10% tomato juice, 0.01% CaCO_3_, and 1.5% agar [38]) at 25 °C in the dark. The bacterial supernatant was prepared from PA culture via careful removal of cell pellets after centrifugation followed by passage through a 0.22 µm sterile filter. *S. aureus* was obtained from the Guangdong Microbial Culture Collection Center (GDMCC). *P. sojae* was obtained from the Institute of Forest Protection of Nanjing Forestry University. All primers used in this study are shown in Appendix A.

### 4.2. Strain Construction

The genomic DNA of the Y4 mutant strain was used as a template to amplify the DNA fragment containing the *Km^R^* cassette flanked by the upstream and downstream sequences of the *sprT* gene. Then, the PCR product was transformed into the FZB42 wild type [10]. The transformants were verified via colony PCR and DNA sequencing.

### 4.3. Transposon TnYLB-1 Insertion Site Mapping

The genomic DNA of the mutants was digested using TaqI. Then, the digested products were ligated using T4 DNA Ligase. The circularized products were then inverse-amplified and sent to Shanghai Biological Company for sequencing. The sequencing results were found by blasting the FZB42 genome to locate the insertion site of the transposon TnYLB-1. The primers used are shown in Appendix A.

### 4.4. Biofilm Formation Test

The bacteria were grown in LB until their optical density at 600 nm (OD_600_) reached 1.0. Then, for the colony surface assay, 2 μL culture was added on the center of a solid LB plate. The plates were dried for 10 min under sterile air in a biosafety cabinet; then, the plates were incubated at 30 °C for 84 h. For the pellicle assay, the cultures were inoculated to 1 mL LB medium contained in a 24-well plate at a 1:100 dilution. After being mixed well, the plates were incubated 25 °C and observed every 12 h. Then, the state of biofilm formation was observed, based on the colony’s ability to produce wrinkles on its surface. Three biological replicates were performed, with each culture being technically replicated in three wells.

### 4.5. Bioinformatics Analysis

The *sprT* gene sequence was inputted in the Uniprot website to obtain basic information. The protein sequences of SprT from different species were downloaded from the NCBI protein database. Unipro UGENE software was used for protein sequence conservation analysis of SprT. SOPMA and SWISS-MODEL were used to analyze and model secondary and tertiary structures of the SprT of FZB42.

### 4.6. Swarming Test

The bacteria were grown in LB until reaching an OD_600_ of ~1.0. Then, 2 μL of the cultures was spotted onto the center of LB containing 0.5% agar (the medium should be used immediately after preparation). After being dried for 30 min in a clean bench, the plates were incubated at 25 °C for 12 h in an incubator with a moisturizing function, and the diameters of the colonies were measured. Three biological replicates were performed, with each culture being technically replicated in three plates.

### 4.7. Sporulation Test

The strains were grown in LB for 24 h, and 1 mL bacterial solution was taken and added to a 2 mL centrifuge tube. The centrifuge tube containing bacterial solution was treated at 85 °C for 10 min. The heat-treated cultures were 5-fold serially diluted. A total of 3 μL of each dilution was spotted on LB plates and then dried for 10 min before being incubated at 37 °C for 24 h. Three biological replicates were performed, with each culture being technically replicated in three spots.

### 4.8. Antagonistic Test of Antibiotic Bacilysin

The method is as follows: Pick fresh FZB42 and Y14 colonies and inoculate them into 10 mL of LB liquid culture. Shake at 37 °C and 200 rpm for 6–8 h. Adjust the OD600 of the pre-culture to 1 and transfer it to 30 mL of PA culture medium at a ratio of 1%. Shake at 37 °C and 200 rpm for 24 h. Transfer the bacterial solution to a 2 mL centrifuge tube and centrifuge at 12,000 rpm for 10 min. Use a syringe to collect the supernatant, filter it through a 0.22 μm filter membrane, and obtain the supernatant containing the antibiotic after 24 h of bacterial growth.

For *S. aureus* as the indicator strain, the culture of *S. aureus* ATCC-9144 at an OD_600_ of ~1.0 was diluted with LB medium at a 1:100 ratio and poured onto LB agar. Holes were made using a sterile punch and filled with 100 μL of bacterial culture supernatants. The plates were incubated at 37 °C for 12 h before the radii of the antagonistic zones were measured. For *S. aureus,* the antagonism radius was defined as the center of the hole to the farthest end of the transparent inhibition zone. For *P. sojae* as the indicator strain, fresh *P. sojae* cake was inoculated onto V8 solid medium. Holes were similarly made and filled with 100 μL supernatants. The plates were incubated at 25 °C for 14 days before the antagonistic effects were observed. The antagonism radius was defined as the nearest distance from the edge of *P. sojae* to the hole center. Three biological replicates were performed, with each culture being technically replicated in three holes.

### 4.9. HPLC-MS

A positive detection mode of HPLC detection of bacterial cultural supernatant was used with 0.1% formic acid added to buffer solution A (H_2_O) and buffer solution B (acetonitrile). Liquid chromatography was performed using a 50 × 2.1 mm 5 μm Hypercarb column with a flow rate of 0.4 mL/min. The sample injection volume was 3 μL, and the LC program was as follows: 0% buffer B for 1 min, linear gradient of buffer B from 0% to 95% within 10 min, linear gradient of buffer B from 95% to 0% within 2 min, re-equilibration with 0% buffer B for 10 min, and then the next sample was loaded. The samples were re-run after dilution with water to avoid saturation of the mass signal detector. The mass spectrometer was set to the following parameters: capillary voltage of 2500 V, drying gas at 350 °C with a flow rate of 10 L/min, nebulizer pressure at 30 psi, and segmented voltage at 125 V.

### 4.10. Real-Time Quantitative PCR

The bacterial cells cultured in PA medium for 24 h were collected for RNA preparation using the Trizol method. Reverse transcription for cDNAs was performed using TAKARA PrimeScript™ RT Master Mix (TaKaRa, Maebashi, Japan), according to the instructions. qPCR was carried out using TAKARA TB Green^®^ Premix Ex Taq™ II (TaKaRa, Maebashi, Japan), according to the instructions. The fluorescence quantitative PCR results were calculated using the 2^−∆∆Ct^ method. The housekeeping gene gyrA was used as the internal standard.

## Figures and Tables

**Figure 1 ijms-24-16815-f001:**
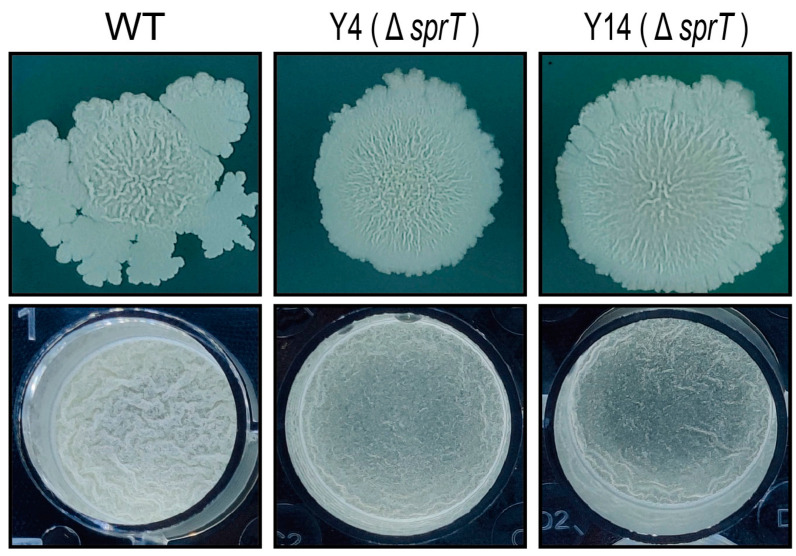
Effect of *sprT* deletion on biofilm formation of *B. velezensis* FZB42. Colonies or pellicles formed by the FZB42 wild type (WT), the *sprT* transposon insertional mutant Y4, and the *sprT* knockout mutant Y14 were evaluated on LB agar at 25 °C for 84 h and in LB liquid medium at 25 °C for 34 h. Each plate represents a representative sample from three replicates.

**Figure 2 ijms-24-16815-f002:**
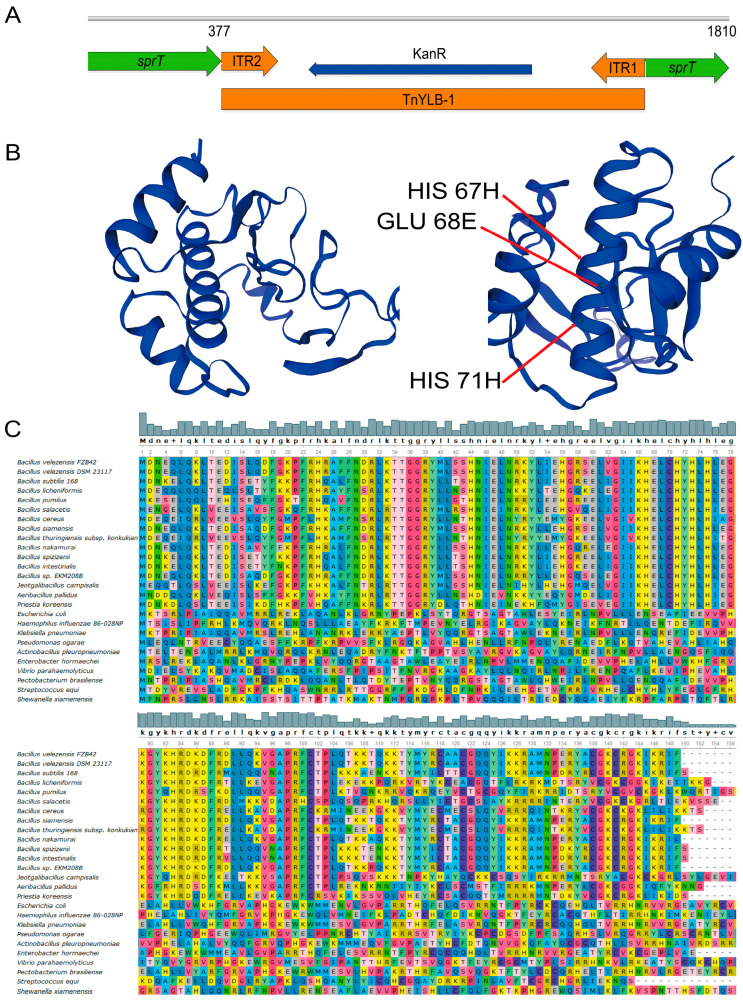
Location of the TnYLB-1 transposon in the mutant Y4 (**A**) and bioinformatic analysis of SprT protein. (**A**) The transposon TnYLB-1 was determined to be inserted between the 377th and 378th nucleotides of the *sprT* gene in Y4. (**B**) The predicted tertiary structure of SprT of *B. velezensis* FZB42. (**C**) Conservation analysis of SprT proteins among different bacteria. Different colors represent different amino acids. The higher the conservation, the more consistent the colors are from top to bottom.

**Figure 3 ijms-24-16815-f003:**
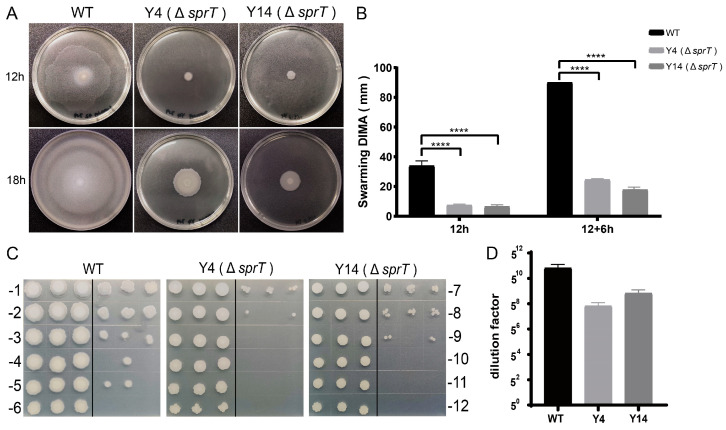
Effects of *sprT* deletion on FZB42 swarming ability and sporulation. (**A**) Swarming of the FZB42 wild type and the *sprT* mutant strains Y4 and Y14 cultured at 25 °C for 12 h (**top**) and 18 h (**bottom**) on 0.5% LB agar. (**B**) Statistics of colony diameters of the WT, Y4, and Y14 after swarming, as described for (**A**). (**C**) Sporulation of the FZB42 wild type and the *sprT* mutant strains Y4 and Y14 cultured at 37 °C on LB solid medium for 24 h. (**D**) Statistics of sporulated cells of the WT, Y4, and Y14 after treatment. One representative plate out of six replicates is shown; T-test was performed, where **** *p* < 0.001.

**Figure 4 ijms-24-16815-f004:**
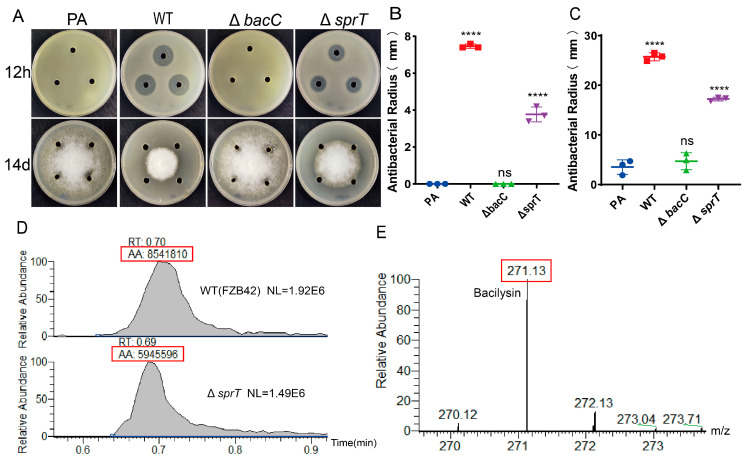
Effect of Δ*sprT* on the bacilysin production of *B. velezensis* FZB42. (**A**) Antagonism of the strains’ culture supernatant toward *S. aureus* ATCC-9144 and *P. sojae*. One representative plate out of three replicates is shown. *S. aureus* ATCC9144 is on the top; *P. sojae* is on the bottom. (**B**) Statistics of antagonism radii of the strains toward *S. aureus* ATCC-9144 incubated overnight (**B**) and *P. sojae* incubated at 25 °C for 14 d (**C**). In panel (**C**), the antagonism radius was defined as the nearest distance from the edge of *P. sojae* to the hole center (mm). (**D**,**E**) HPLC-MS analysis of bacilysin produced by the FZB42 wild type and Δ*sprT*. T-test was performed, where **** *p* < 0.001.

**Figure 5 ijms-24-16815-f005:**
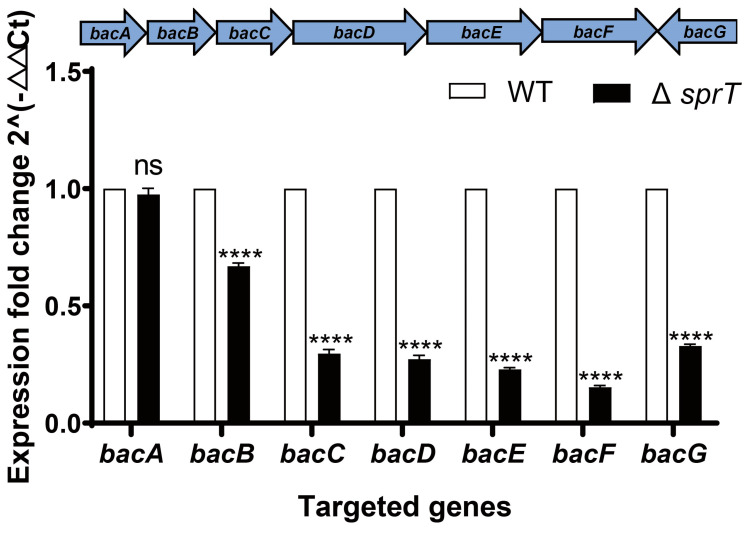
Relative mRNA levels of bacilysin biosynthetic genes in the FZB42 WT and the Δ*sprT* mutant revealed via qPCR. RNA was extracted from bacterial cultures in PA medium 24 h after inoculation for qPCR. The expression fold change was normalized to the expression of the housekeeping gene *gyrA* in WT. Three biological replicates and three technical replicates were used for each gene. T-test was performed where **** *p* < 0.001.

## Data Availability

Data are contained within the article and Appendix A.

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
