# Peer review of "The sprT Gene of Bacillus velezensis FZB42 Is Involved in Biofilm Formation and Bacilysin Production"

_ijms, 2023, doi:10.3390/ijms242316815_

Round 1
Reviewer 1 Report
Comments and Suggestions for Authors
The paper focuses on a very relevant issue which is the “Bacillus velezensis FZB42, a representative strain of plant growth-promoting rhizobacteria 10 (PGPR), can form robust biofilm and produce multiple antibiotics against a wild range of phytopathogens. In this study, we observed different biofilm morphology of the mutant Y4, derived from 12 a TnYLB-1 transposon insertion library of B. velezensis FZB42. We identified that the transposon was inserted into the sprT gene in Y4”
Recent decades have seen a rise in environmental awareness and an interest in more sustainable agricultural production systems. A safe alternative to agrochemicals is the use of PGPR (plant growth-promoting rhizobacteria), which has great potential to increase the productivity of agricultural crops exposed to environmental stress and nutrient deficiency, e.g., through the mobilisation of nutrients and the synthesis of various metabolites (phytohormones, siderophores) and many other features.
The paper deals with important issues:
· 2.1. sprT is involved in biofilm formation of B. velezensis FZB42
· 2.2. Bioinformatic analysis shows sprT is conserved among Bacillus species
· 2.3. sprT deletion resulted in a defect in swarming ability and sporulation of FZB42.
· 2.4. sprT deletion reduces production of the antibiotic bacilysin.
· 2.5. Transcription of the bac operon is down-regulated by ΔsprT deletion
The authors provided interesting results that showed that have demonstrated the importance of sprT for biofilm for-208 mation, swarming, sporulation and, antibiotic production in FZB42. Understanding the 209 role of this metalloprotease in the synthesis of FZB42 antibiotics can help use the PGPR 210 strains such as FZB42 to play a better role in biocontrol. In the future, in-depth studies can 211 be performed to uncover the molecular mechanisms underlying the effects.
The article presents in a legible and transparent manner the material and methods used in a given research work. The methodology is clear and described concisely. Introduction section is comprehensive and is also written in an concise and clear manner. The literature is well-chosen and the conclusions clearly refer to the conducted research.
There are, however some minor weaknesses in discussion, therefore I recommend minor revision of the paper.
Minor issues to be corrected:
· Introduction
It would be good to add a citation to the statement (line 36 and line 45).
· Discussion
I suggest to extend the literature citation on “With a pyramid of previous studies, B. velezensis FZB42 has become one of the most 162 well-studied PGPR strains” (line 162 - 164).
Conclusions could be a bit more developed – especially extracted from the own results, not only the broader but not detailed statements
In summary, the paper is worth publishing in the Journal.
Reviewer 2 Report
Comments and Suggestions for Authors
Bacillus velezensis FZB42 is the model strain for Gram-positive plant-growth-promoting and biocontrol rhizobacteria, which has been the subject of more than 100 published articles in the last 20 years. The current manuscript is devoted to further previously unknown aspects of this bacterium. The results obtained are very interesting. The authors determined, among others, the role of sprT in biofilm formation, motility, ‘endospore development’, and antibiotic production of B. velezensis FZB42. Manuscript should be published in IJMS after making some additions.
Remarks
Line 50 sporulation - it should be explained what process exactly is involved and what does it matter whether they are produced in greater or lesser numbers (e.g. depending on sprT). Most bacteria do not produce spores like fungi, which can be referred to as sporulation. In Bacillus, these are probably endospores, which have a different character than typical spores in fungi.
Line 49-50 The production of these antibiotics in B. velezensis FZB42 is well connected with other phenotypes such as biofilm formation, motility, and sporulation – relevant literature should be cited
Line 140, Line 142 it should be P. sojae
Line 167 it should be Bacillus spp. (spp. not italic, and one dot)
Line 214 Chapter 4.1 It is necessary to provide the origin of the Staphylococcus aureus and Phytophthora sojae strains. How were they identified?
Figure 2C requires further detailed explanation
Gigure 4A requires further explanation (where is S. aureus and where is P. sojae) so that there is no need to guess
Line 142-143 Antagonism radius – it should be defined in Materials & Methods
Chapter 4.4 The methodology for biofilm formation is not complete. Further procedure should be provided. The same applies to chapters 4.6 and 4.7. There are also no details regarding inhibition zone measurements (Chapter 4.8). Much of the data that should be included in the methodology is found in Results.
Line 252 Sporulation test - the methodology must be completed, it is as if there was no completion of how to proceed to determine the sporulation
Line 347 the authors should be corrected in accordance with the requirements of the Editorial Office
